# eHealth Engagement on Facebook during COVID-19: Simplistic Computational Data Analysis

**DOI:** 10.3390/ijerph19084615

**Published:** 2022-04-12

**Authors:** Caroll Hermann, Melanie Govender

**Affiliations:** Department of Psychology, University of Zululand, KwaDlangezwa Campus, Richards Bay 3886, South Africa; govenderm@unizulu.ac.za

**Keywords:** social media, netnography, mental health, natural language processing, visualization, data analysis, COVID-19

## Abstract

Understanding social media networks and group interactions is crucial to the advancement of linguistic and cultural behavior. This includes how people accessed advice on health during COVID-19 lockdown. Some people turned to social media to access information on health when other routes were curtailed by isolation rules, particularly among older generations. Facebook public pages, groups and verified profiles using keywords “senior citizen health”, “older generations”, and “healthy living” were analyzed over a 12-month period to examine engagement with social media promoting good mental health. CrowdTangle was used to source status updates, photo and video sharing information in the English language, which resulted in an initial 116,321 posts and 6,462,065 interactions. Data analysis and visualization were used to explore large datasets, including natural language processing for “message” content discovery, word frequency and correlational analysis as well as co-word clustering. Preliminary results indicate strong links to healthy aging information shared on social media, which showed correlations to global daily confirmed cases and daily deaths. The results can identify public concerns early on and address mental health issues among senior citizens on Facebook.

## 1. Introduction

This study investigated how social media users viewed health advice on Facebook pages and groups from 1 October 2020 to 30 September 2021 during the global COVID-19 pandemic. The aim of this study was to determine how users engaged social media, their sentiments, and if any themes were detectable from the content of messages. While social media studies may still be in its infancy, it should be considered for research as it contains the richness of large datasets with content from large groups and pages delivering information on health issues.

Fluid social media may be defined as centers for social groups (or community) or networks that initiate conversations and enable relationships to form [1]. Social media may also take on a creative role or the role of information sharing [1,2]. Freberg [3] (p. 772) claimed that social media are used to “engage, reach, persuade and target specific audiences across multiple platforms”. Social media data can also provide insight into concurrent behaviors within communities [1], and traditional methods in research enable testing and evaluating (p. 54) the substantial unstructured data obtained online. Freberg [1] also emphasized growing expectations to make sense of online data. This study investigated how people engage in healthy and active aging activities through participation in Facebook groups, pages and verified profiles through reactions, shares and follows, etc., during a period of globally restricted movement. Facebook, presumably the largest social media platform [1], was selected as the platform to investigate how senior citizen’s users engage health information on pages and groups during the COVID-19 lockdown from October 2020 to September 2021. Facebook claimed to have had 2.9 billion active users, of which CrowdTangle (a subsidiary of Meta who also owns Facebook) could track public content of over 7 million Facebook pages, and groups [4].

Users can create personal profiles and join virtual interest or hobbies groups and through these interactions, researchers can access valuable datasets [5,6]. Due to the ease of accessibility through CrowdTangle, Facebook was selected over other social media platforms for this study. Instagram (short messages and pictures) and Reddit (missing important data, such as followers) were excluded as they provide limited information compared to Facebook, such as “Post types” and “Top Admin Country”. Twitter was excluded because access was limited to the most recent seven days’ tweets primarily through CrowdTangle [4]. 

Since the advent of social media (Facebook, YouTube, Twitter, TikTok, etc.), there is an increased reliance among people on social media and the internet to keep informed on current affairs, issues of health, social activities, hobbies, and various other aspects of interest. A recent study in Canada found that approximately 80% of social network users are aged between 15 and 34 years old and rely exclusively on social media for news and current affairs [7]. In 2021, a report stated that 70% of Americans use social media in any form “to connect, share information and for self-entertainment” [8]. According to this study, there has been a steady decline in participants in the 18–29-year-old age group since 2019, while the number of participants in the 30–49-year-old age group has stagnated. However, the number of participants in the 65 years and over age group has shown steady growth, a 30% increase, since 2012. While Facebook is popular among all age groups, it appears to be more popular among adults [8]. In a survey in 2014/2015, Facebook was the most popular social media platform among American teens, which has since dropped from 71% to 51% users in the 13–17-year-olds age group [7], while 36% of adults stated that they read news on Facebook.

Facebook was designed to enable maximum spread of the network [8] and has become a prime tool for businesses and researchers. A study performed in 2017 reported that 84% of the Fortune 500 companies have a Facebook page [9]. Social media platforms are often used by groups for a specific community (such as “Healthy Living Group”) whose main purpose is to educate. By using social media strategically, aiming content at a particular audience, much information can be disseminated as well as collected [1]. Audiences, relationships, personalities, content, actions and innovativeness became very important during COVID-19 lockdown globally [1]. 

Posts, predominantly health-related posts, became cardinally important during COVID-19 lockdown. During the isolation stages of the global pandemic, social media were the primary source of communication. Globally, online activity increased during the pandemic [10]. Such information may provide an opportunity for a valuable field of new research which uses social media data through third-party software or Application Programming Interface (API).

Facebook reactions include “Like”, “Wow”, “Love”, and “Haha” (grouped as “Total Positive” reactions) and “Sad” or “Angry” (grouped as “Total Negative” reactions). There has always been a question of how much credibility one can assign to “Likes” [7,9] and La France [11] claims that reactions were designed to “blunt-force emotional reaction” (p.19), by the click of a button and Lee et al. [12] feel that “Likes” can drive traffic and serve a socialization component, but admit that the connotation can have variations. These reactions, including “sharing” behavior, pose social risks. Praet et al. [13] feel that reactions on social media serve to be mere “echo chambers” (p. 3). 

Netnography also evolved during the pandemic. Netnography gives users a lens or an opportunity to address cultural, health or other burning issues and, when applied to the common good, can facilitate change [14]. Netnography studies the online interactions between individuals through internet connections or computer-mediated communications [15]. These communications bind the user with more than just transmission of information, they also tie people through a “common interactional format, location or ‘space’,… or virtual ‘cyberspace’” (p. 16). Ellison et al. [5] claim that there is a correlation between online and offline behavior. Online social networking supports maintenance of existing social ties and the formation of new ones and builds social capital such as shared interest. 

Social engagement has mainly been defined by marketers and can be categorized into four distinct levels: connection, interaction, loyalty and advocacy [16]. Shawky [16] claims that when new people join (connection, and lowest level of engagement), all are encouraged to contribute (interaction) and take part over a long period (loyalty) as well as to share and promote among other networks (advocacy, also the highest level of engagement). 

This manuscript investigated older generations’ level of health-seeking behavior on Facebook related to active and healthy aging posts. The term “active aging” is considered a process “optimizing opportunities for health, participation and security in order to enhance quality of life and well-being” [17]. Participation includes active involvement in the labor force, be it socially, economically, culturally or spiritually. Time spent on social media significantly increased during the global pandemic, as most people work from home and spend their time online. Most countries have policies to encourage people to work as they grow older and to delay chronic illnesses that are costly to the state and health care systems.

Data visualization and storytelling were used to illustrate how followers of groups and pages accessed Facebook to seek information. Often, data are digitalized to further analyze data and trends [18,19]. Data visualization and effective communication thereof can turn insights into action [20]. Recently, digital storytelling gained popularity among researchers in the Humanities. Communicating information using storytelling forms a crucial instrument for effective communication [10]. 

## 2. Materials and Methods

This study used computational sciences to analyze the data, evaluated results quantitatively and engaged with digital humanities. One of the key challenges for digital humanities is the integration of critical and technological engagement. Svensson [21] claimed that “early career scholars are coming to the field…. because it is seen as a place to engage with the future of humanities.” (p. 1). 

Two strategies are employed when collecting online data: systematic collection via various platforms or third-party software; systematic organization and cleaning of raw data, and the analysis of organized data [1]. This study utilized CrowdTangle (a third-party content discovery software) to collect social media performance on Facebook groups and pages.

CrowdTangle captured the search words “senior citizen health”, “older generation” and “healthy living”, which resulted in 6,462,065 interactions with 116,321 posts on public Facebook pages and groups in the English language from 1 October 2020 to 30 September 2021. COVID-19 data were obtained from the Johns Hopkins University Dashboard [22]. 

The Facebook groups search resulted in 3113 interactions stemming from 294 posts, and the Facebook pages search resulted in 62 758 interactions from 896 posts. The second phase of handling data involved cleaning CrowdTangle.csv files. The corpus was prepared using Excel spreadsheets combining social media and COVID-19 [15] data from 1 October 2020 to 30 September 2021. The three sets of data extracted from the combined set were: (i) a combination of COVID-19 statistics and Facebook “Likes at Posting” and “Followers at Posting”, (ii) Facebook pages and groups cleaned and (iii) “message” to allow for easier computations. Preprocessing included removal of columns in the Facebook data that were deemed unnecessary, such as identifiable information—“User Name”, “Facebook ID”, “Page Description”, “Date Page Created”, “Post Created with Date and Time” (only “Post Created Date” was retained and name changed to “Date”), “Post Created Time”, “Video Share Status”, “Is Video Owner?”, “Post Views”, “Total Views”, “Total Views for All Cross Posts”, “Video Length”, “URL”, “Link”, “Final Link”, “Image Text”, “Description”, “Sponsor ID”, “Sponsor Name”, and “Sponsor Category”. OpenRefine 3.4.1 was used to remove trailing and leading white spaces and was further prepared by simplifying column names and removing duplicated columns, such as “Date” in both datasets. Data were filtered according to page categories and non-health categories were removed and cleaned of spam and advertisements—27,899 entries remained. All text in messages were transformed to lowercase, white spaces removed and transformed to “text” using OpenRefine. All “NaN” and missing values were removed. 

### Analyses

The combined dataset was cleaned using OpenRefine (Open source project), and Python 3 was used for content analysis, visualization and data analyses in most cases, using Matplotlib(The Python Software Foundation, Delaware, United States of America), Seaborn (pandas library for creating graphs in Python), Plotly (pandas libraries for creating graphs in Python) and TextBlob (pandas library used in simplified text processing). Both cleaned datasets were imported to a workbook in Jupyter Notebook (open source computing programme). 

Mean, standard deviation, distribution and correlations were obtained from both datasets. Summaries of total COVID-19 cases, daily rates, cumulative deaths, daily deaths, likes and followers of posting, types of posts, page administrators’ country of origin and page categories were plotted to provide a visual overview of data. A histogram and probability were plotted against new cases. Page and group administrators’ top 10 countries were determined, and top 10 page categories and types were differentiated. The top 10 pages with the most posts were also identified. Messages (which included status updates and titles of photos or videos) were analyzed for sentiment polarity, message length, word count, top 10 bigram and trigram words were identified, excluding stop words. Words were tagged and analyzed, as well as bivariate analysis performed. Finally, topic modeling was performed and presented schematically using Latent Dirichlet Allocation (LDA) in GenSim (open source software designed to handle large text collections). 

Ethical mining of social media protocols was strictly adhered to. No human subjects were involved in the collection of data. This study was registered with the University of Zululand Research Ethics Committee (UZREC 171110-030 Dept. 2021/1) and all identifying information removed from the raw dataset. 

## 3. Results

### 3.1. Descriptive Analysis

Of the total data obtained from CrowdTangle [23], 696 unique posts were identified. The concatenated dataset consisted of Facebook pages and groups and WHO COVID-19 Daywise (daily infection rates) [22] numbers, resulting in 28,948 rows and 28 columns of data. 

Dataset 1 [11] contained 365 (from 1 October 2020 to 30 September 2021) rows of information in 6 columns, arranged by Confirmed COVID-19 Cases, Confirmed Deaths, New Cases, News Deaths, and Number of Countries. 

Figure 1 depicts the trends of new COVID-19 cases and deaths grouped by month over a 12-month period and there is a clear relationship between deaths and infections. As infections rose, so did deaths.

Table 1 summarizes the descriptive statistics of Datasets 1 and 2. Dataset 1 was based on the cumulative infections, deaths and daily new infections and deaths. 

Dataset 2 [12] contained 28,948 rows of information in 21 columns of clean data. Table 2 depicts the descriptive statistics of Dataset 2 based on Facebook pages and groups over a 12-month period on health and aging. The dataset describes the number of likes and Followers at Posting, and other reactions, such as love and wow emoticons. The “Total Positive” includes all the positive reactions, excluding “Sad” and “Angry”, as these were grouped under “Total Negative”.

The researchers attempted to determine whether a noticeable trend existed between “Followers at Posting”, “Likes at Posting”, the number of “Comments” made on posts, as well as the “Total Positive” and Total Negative” reactions based on the fluctuating COVID-19 daily figures. Based on Figure 2 below, there is little evidence of a relationship between these variables.

Reactions on health posts increased slowly, whereas negative sentiments decreased over time, as shown in Figure 3.

As new cases of infection around the globe fluctuated, so did the social engagement increase, as shown in Figure 4.

Facebook allows posts as status updates, links, photos, native videos, live videos complete and scheduled, videos and YouTube clips as posts. Figure 5 below indicates that photo posts were in the majority (18,722) and scheduled live videos (42) were the least.

Most pages and groups had administrators that originated from specific countries. Some 128 countries were represented in the dataset, with the USA having the most administrators (16,247 or 27%), and India (1976 or 21%) and Canada (1515 or 18%) in the top 3. Figure 6 represents the top 10 countries represented by page and group administrators. 

When creating pages and groups, administrators can assign the pages to specific categories (Figure 7) in order for members or followers to find them. There were 180 unique categories with the category “Gym” (6028) having the most posts, followed by “Media News Companies” (2747) and “Health Site” (2742) posting the second- and third-most posts in a 12 month period. 

### 3.2. Page Names

A total of 10,769 unique page names occurred in a 12-month period, showing that several pages posted more than once. The top 3 pages or groups posted 779,545 and 261 times, respectively. In most cases, the page name contained the word “health” in it, as displayed in Figure 8 below.

### 3.3. Categorical Analysis

Information was ordered according to the top 12 page categories, namely: ‘HEALTH_SITE’,’HOSPITAL’,’GYM’, ‘UNIVERSITY’, ‘RETIREMENT_ASSISTED_LIVING_FACILITY’, ‘MEDICAL_HEALTH’, ‘ACTIVITY_GENERAL’, ‘HEALTH_BEAUTY’, ‘COMMUNITY_ORGANIZATION’, ‘COMMUNITY’, ‘PERSON’, and ‘HEALTH_SITE’,’MEDIA_NEWS_COMPANY’, as depicted in Figure 9 below.

### 3.4. Message

The “messages” retrieved were analyzed using Natural Language Processes in Python, using TextBlob and took 1.9 s to run. Messages were analyzed for word, character, and sentence count, including average word and sentence length, which is depicted in Table 2 below. 

Messages were analyzed for sentiment and polarity as depicted in Figure 10 below. Posts regarding aging and health were overwhelmingly in the mean positive sentiment at 0.27. Mean message length was at 650 characters with a minimum of 2 to maximum of 5660. Posts with status updates and photos, videos and links with descriptions contained a minimum of 1 word, a maximum of 1038, with a mean of 98.12.

Word embedding or vectorizing is an important tool for understanding the context of words in NLP [24]. The concept refers to the meaning of words in relation to their distribution in the text. In Figure 11, the word being investigated is represented by a red dot and relates to other grammatically similar words. Representations of input words form an important part of NLP research [20]. In the example below, the word “health” appeared in both the top bigram and trigram words. The word vector represents probability distributions of the word “health”. 

## 4. Discussion

Two datasets were investigated to identify any relation between social media health posts and the various COVID-19 new infections and or new deaths over a 12-month period. The relation, as depicted in Figure 1, between daily new infections and daily new deaths was identified in various research documents, and thus corroborates this trend [25,26,27,28]. 

A pivotal note is that the amount of “Likes at Posting” (and even “Total Positive”) is consistently higher than the amount of “Shares” contrary to the assumption that if someone “liked” a post, they would want to disseminate that information. Being mindful of instances of “echo chambers” [11,28] and “emotional blunting” [10,29], Figure 2 indicates Facebook reactions responding to the peaks and valleys of the infection rates, except in one instance in August 2021. The content of the posts was not investigated, but one could assume that the online (trend) behavior would influence offline behaviors, and page followers and group members would lead healthier lives [5], as was stated in earlier research—that the directionality of behavior was online to offline.

Connectedness plays a big role in the evaluation of effectiveness. The “Likes” indicates the followers’ involvement with the group and can have a huge impact on the programs [16]. Figure 2 indicates high connectedness on “Likes” and “Total Positive Reaction”. According to Shawky [16], connectedness is the lowest level of interaction, but it also leads to communicating with existing followers and reaching new audiences. Furthermore, “Likes at Posting” and “Followers at Posting” follow a similar trend, indicating close correlation. Further analysis of the same dataset, focusing on content analysis, will investigate deeper significance of the results.

The progressive trend line in positive reactions could be attributed to resilience strategies by re-engaging in gratifying past activities such as exercising. Increased emphasis was placed on the importance of physical health and, by implication, mental health [25,30]. To promote healthy behavior, pages and groups employed several strategies, such as ‘getting out of your fans’ way [31], sharing expertise through texts and visual posts. The health pages and groups in the sample focused on positive posts to improve physical health (Figure 3 and Figure 10). 

Evidence in Figure 3 indicated an increase in positive reactions and a decline in negative comments within the period of a year. Founded on various studies [12,13], this may be attributed to a blunting or desensitization effect. It may, however, be more realistically attributed to a more general concern for physical and mental well-being [30], and therefore an increased awareness of healthy ways of living. 

Figure 4 shows a rise in social engagement on Facebook pages and groups as Facebook also reported a steady increase in users for the same period [32]. According to Dykes [17], social media is increasingly used to disseminate information and facilitate change [28]. In this 2021 study, Osuwu-Ansah et al. [33] claim that groups and pages serve the “purpose of information-sharing, peer-tutoring, learning and finding friends” (p. 7), and also state that social media competence can be a hindrance. Older generations might use Facebook to “learn” from and “share” only, rather than comment or voice their opinion. 

Most posts (Figure 5) on aging and health were photographs, followed by links. Photographs are one of the most popular features on Facebook. Personal photos can be shared, tagged and, when provisioned, have their own comment sections, which allow for conversation. Often, campaigns encourage followers to tag or share photos [31]. It is relatively easier to share an existing post than to “make up” new text posts, which could be explained by the “echo chamber” phenomenon [11]. The extent to which followers engage with posts is not within the scope of this research and remains debatable. 

As expected, 27% (Figure 6) of the administrators of pages and groups that posted on health and aging originate from the United States of America. Northern America is purported to have the third-largest population that use the internet [34], of which the USA and China are the highest users of social media with a reported 71.5% Facebook subscribers in the USA [35].

Most categories that pages belong to are institutions or companies which promote health, such as groups or pages within the categories of “Gym”, “Health Site”, and “Hospital”. People who search for health information or join groups with health topics would be drawn to these categories and to pages with “health” in the name.

Most page or group names in Figure 8 have the words (or stems of the words) “health” or “family”. Names of groups or pages reflect the content and will inevitably rank higher on a search item. The pages or groups will have similar names. 

Although thematic/content analysis was not in the scope of this study, a cursory categorical analysis revealed that pages in the “Gym” category (Figure 9) included the most posts on healthy aging and/or contained posts aimed at older members. Content analysis will give further details on the “types” of posts and updates on these health sites, such as information, misinformation or information for fun, phatic posts or positive trends [5]. 

Healthy living or actions that would improve your health, in essence, are a positive sentiment. This is corroborated by the majority of posts (Table 2 and Figure 10) having a positive sentiment. Most members responded positively to posts, except on two occasions (Figure 10). Content analysis will allow for further investigation of these posts and determine whether there was a related trend in COVID-19 cases.

As shown in Figure 11, words such as welfare, services, patients, doctors, prevention, Medicare, education, and social are used either before or after the word “health” on the post mentions. Much emphasis is placed on “aids”, “benefits”, “prevention”, “nutrition”, “social”, etc., in the words used on Facebook. These words have positive connotations and it is expected that users reacted positively to these posts by liking, sharing or reacting with emoticons, such as “Like”, “Love” and “Caring”.

This study is based on a large dataset which would have otherwise been difficult to obtain and therefore places importance on datasets obtained from third-party software. Most studies in the Humanities and in Psychology focus on the narratives (in messages), but few focus on the statistical analysis of such narratives. This study fills a gap in social transmission of knowledge and can be useful when attempting to grasp cultural dynamics on social media. Health topics remain popular within all social network sites and assist researchers in better understanding of social media and their users [36]. This study provides insight into pages and groups aimed at more senior citizens’ use of health information on Facebook.

Few studies in the Humanities and the Social Sciences have used social media data and even fewer still made use of computational sciences for analysis [5]. It is difficult to analyze big datasets with traditional statistical or language processing software and computational sciences afford researchers the opportunity to work with big datasets. Most research currently focuses on politicking and false information, whereas the focus of this study was an investigation of online behavior, tracking likes, followers, and sharing of health information. A further study is being conducted on the content of the messages. Further options for research would be to include several social media sites to compare information.

There were several limitations to this study. Only one social media platform was analyzed (Facebook) and this could contribute to sample selection bias. Facebook covers a broad spectrum of any population (assumed older population) [37] due to its popularity, whereas other platforms, such as Instagram and Twitter, cover specific subgroups, such as younger people. Facebook data were therefore deemed sufficient for the purposes of this study. This opens an opportunity for further study to include other social network sites, such as Twitter, LinkedIn, and YouTube.

This study further highlights the extent to which older users on Facebook relied on health information during the era of the global pandemic and how administrators of health sites could or made use of promoting healthy lifestyles or living. Research during the current global pandemic should adapt and innovate [14] and social media content is an untapped resource begging for discovery.

It was not the aim of this study to investigate the credibility of the information on Facebook, nor identify misinformation, but to investigate how and to what extent information was disseminated. The “types” of posts, such as photos or videos, were not investigated; only “statuses” were analyzed and it would be interesting to include other types of posts in future studies. This provides leeway for further training in digital data analysis and more in-depth thematic analyses, such as topic modeling and natural language processing (NLP) [33]. The focus of this study was not on the influence of COVID-19 posts, but on statuses that focused on health during the specific COVID-19 period.

Specific thematic/content analysis was not within the scope of this research and therefore peak-and-valley incidents cannot be explained. The content of the posts was not analyzed at this stage and provides scope for further studies. Currently, natural language processing (NLP) is being employed to analyze the same “message” category for content analyses. Pathological use and rumination on health conditions were not investigated and do not form part of this study. 

## 5. Conclusions

This study attempted to visualize data obtained from Facebook through CrowdTangle. This study aimed to show links between status updates during the COVID-19 pandemic and health information-seeking behavior in older generations. This study also looked at using simple coding to make this analysis and how it would present information. This study shows that positive information-seeking behavior increased during COVID-19 lockdown. This information may influence how information is disseminated in the future, and how health information sites can influence behavior. Organizations may find the results useful to fine tune communication and marketing strategies. Additional attention should focus on users’ reactions to posts and updates. Social media may not be the sole playground of the young and the youthful, as it caters for all ages, across diverse spectrums, but such demographic data are not obtainable through CrowdTangle. It is envisaged that this study will also encourage more research on social media usage, with the utilization of large and abounding datasets.

## Figures and Tables

**Figure 1 ijerph-19-04615-f001:**
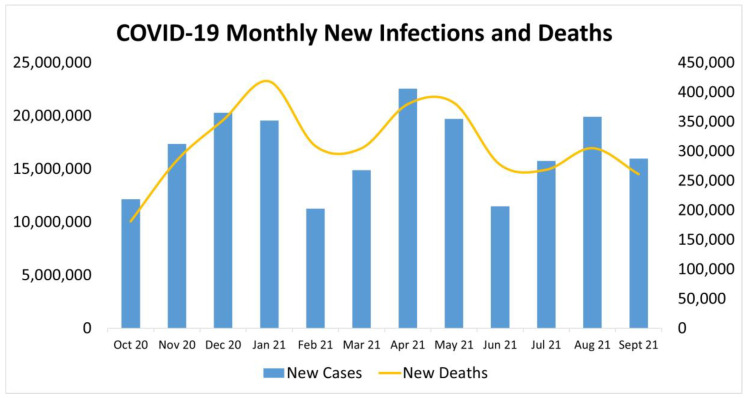
COVID-19 monthly new infections and deaths. Graphic presentation of monthly new infections of COVID-19 and monthly new deaths from October 2020 to September 2021.

**Figure 2 ijerph-19-04615-f002:**
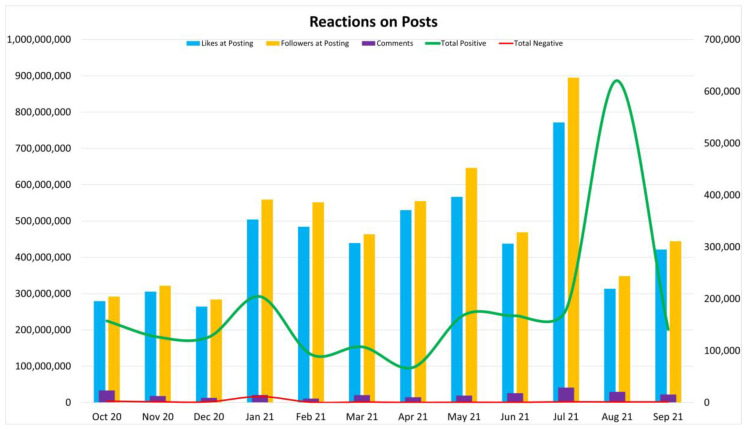
Reactions on posts. Digital representation of reactions on messages on Facebook pages and groups.

**Figure 3 ijerph-19-04615-f003:**
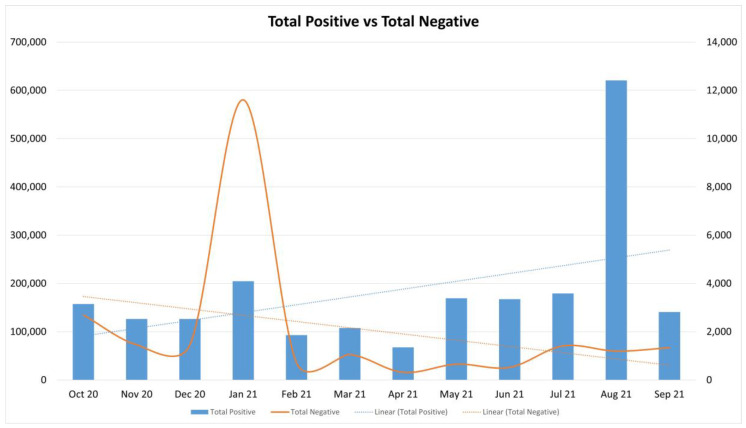
Total Positive reactions vs. Total Negative reactions on Facebook page and group message posts with trend lines.

**Figure 4 ijerph-19-04615-f004:**
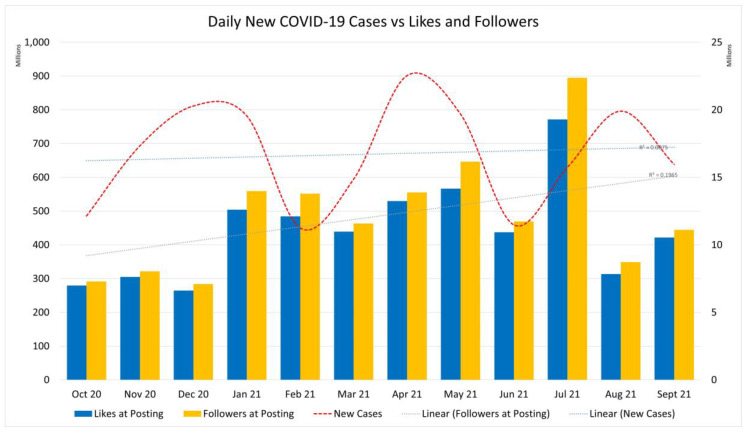
Daily new COVID-19 cases vs. likes and followers. Depiction of new cases, likes and followers at posting with trend line.

**Figure 5 ijerph-19-04615-f005:**
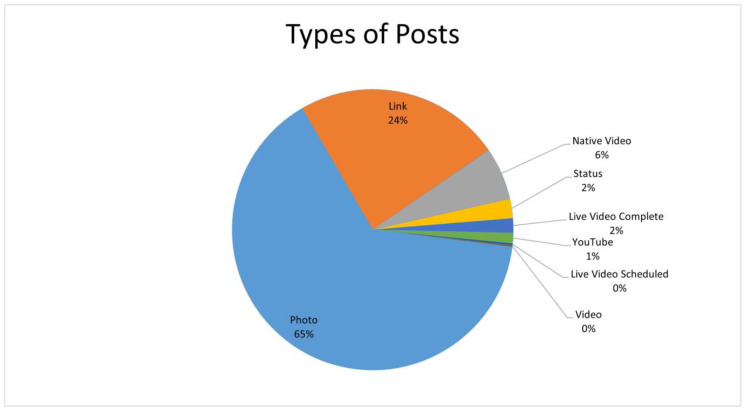
Types of posts on Facebook pages and groups.

**Figure 6 ijerph-19-04615-f006:**
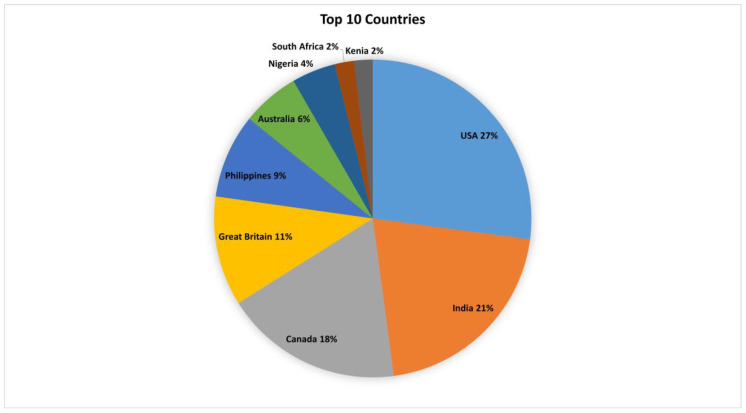
Top 10 countries. Page administrators from the top 10 countries of origin.

**Figure 7 ijerph-19-04615-f007:**
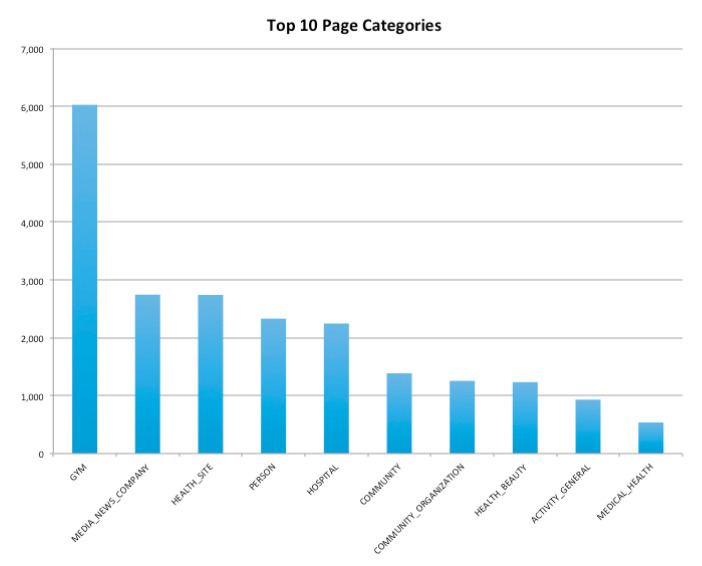
Top 10 page categories. Representation of top 12 page categories on Facebook pages and groups.

**Figure 8 ijerph-19-04615-f008:**
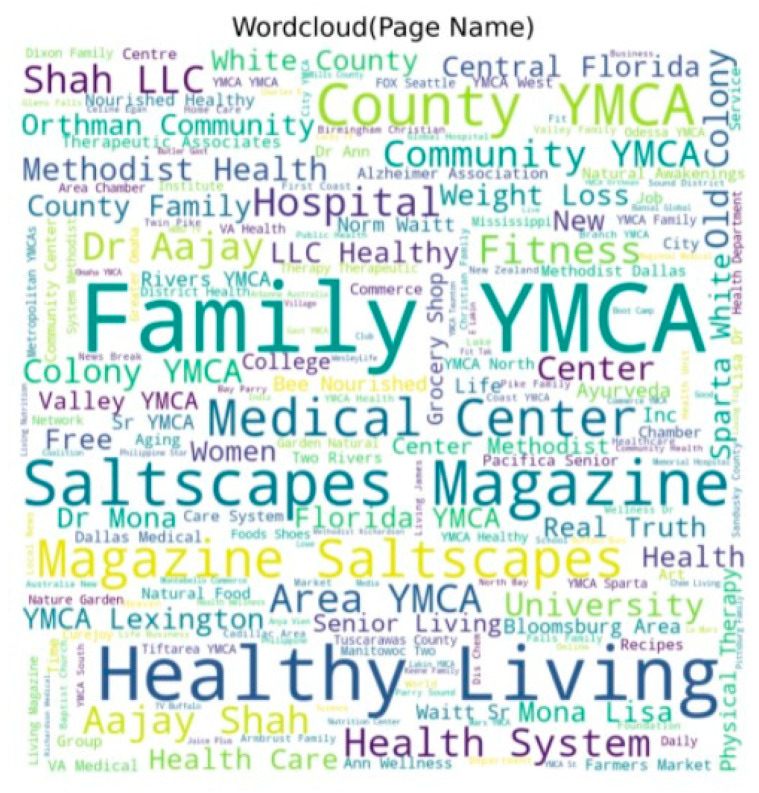
Page names. Word cloud representation of names of Facebook pages and groups.

**Figure 9 ijerph-19-04615-f009:**
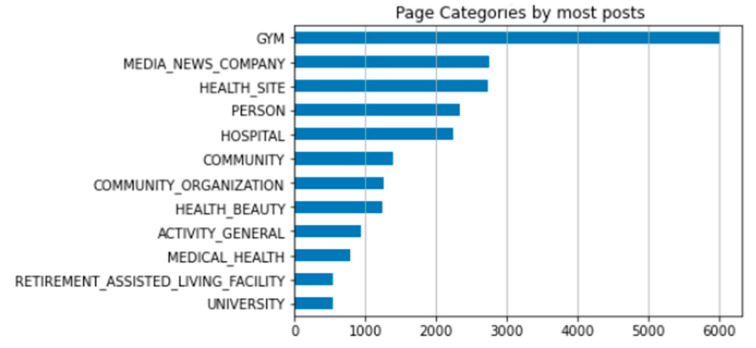
Graphical representation of most posts per page category.

**Figure 10 ijerph-19-04615-f010:**
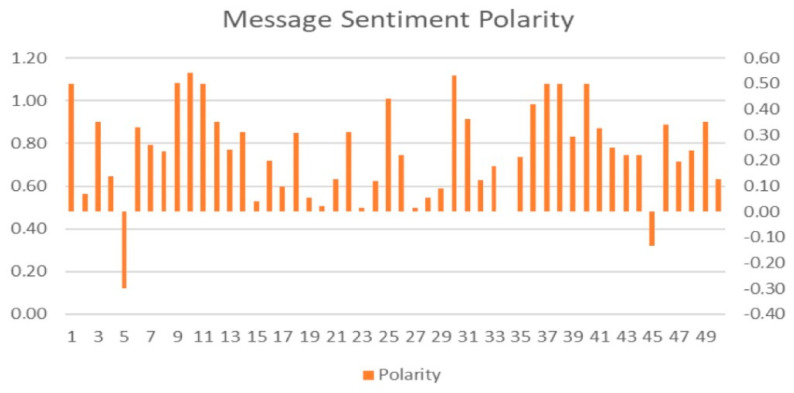
Message sentiment polarity. Visualization of sentiment polarity of messages.

**Figure 11 ijerph-19-04615-f011:**
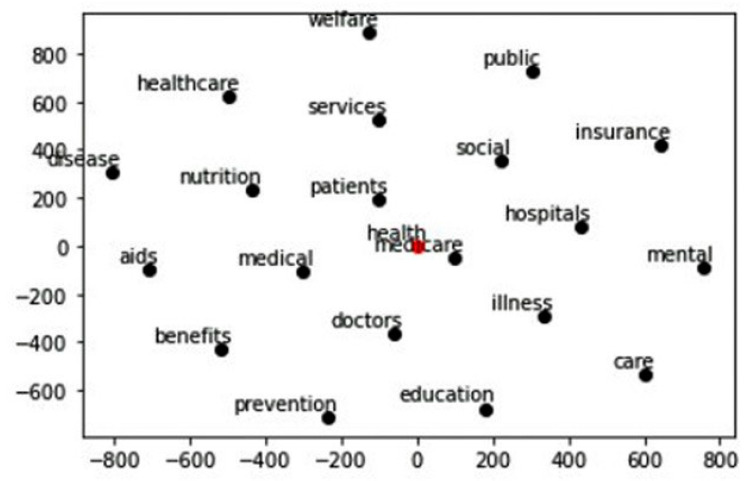
Health: probability analysis. Schematic representation of the probability distribution of the word “health” which is represented by a “red” dot in the drawing above.

**Table 1 ijerph-19-04615-t001:** Combined Datasets 1 and 2. Descriptive summary analysis of COVID-19 global statistics—Datasets 1 and 2.

Description	Descriptive Statistics				
	Total	Mean	SD	Maximum	Minimum
Cumulative Infections	48,674,562,820	133,354,966.6	58,988,027.8	23,344,208,971	34,330,972
Cumulative Deaths	1,067,685,110	29,225,164.7	1,145,757.8	4,784,857	1,073,081
New Infections	200,740,517	549974	156,158.1	1,503,571	260,298
New Deaths	3,720,389	10 192.8	2991.8	20,759	3630
Likes at Posting	5,317,206,337.00	206,279.00	19,336,889.6	117,508,790.00	2.00
Followers at Posting	5,829,532,135.00	227 597.2	2,225,409.00	117,750,435.00	2.00
Likes	1,815,171.00	63.1	2863.4	4,755,906.00	0.00
Comments	187,810.00	6.6	93.9	7,088.00	0.00
Shares	335,878.00	11.7	208.2	24,162.00	0.00
Love	253,075.00	8.9	146.9	12,376.00	0.00
Wow	20,730.00	0.7	24.2	2725.00	0.00
Haha	107,333.00	3.7	177.4	25,048.00	0.00
Sad	17,452.00	0.6	30.2	3,849.00	0.00
Angry	6301.00	0.2	11.5	1,816.00	0.00
Care	43,446.00	1.5	77.1	10,193.00	0.00
Total Positive	2,132,422.00	74.2	2929.2	482,661.00	0.00
Total Negative	23,753.00	0.8	35.3	3,893.00	0.00

**Table 2 ijerph-19-04615-t002:** Summary of descriptive analysis of messages. Examples of messages, sentiment polarity, message length, word length and average word length.

ID	Message	Polarity	Message Length	Word Count	Average Word Length
0	Area Agency on Aging is providing Healthy Livi…	0.5	158	19	7.37
1	?? The Not Old - Better Show || Episode #484 "…	0.07	790	136	4.82
2	Aging & Long Term Care of Eastern Washington i…	0.35	468	71	5.61
3	Come join the New Womens Movement. Also, leave…	0.14	132	27	3.93
4	Reishi can help slow your aging, boost your im…	−0.3	392	32	11.28
5	Virtual Brain Bus: Healthy Living for Your Bra…	0.33	1380	214	5.45
6	PITCH 3?? ??Hello everyone, I am Gemma and I r…	0.26	1769	294	5.02
7	??????BUZZY BEE?????? Healthy Living?Try our P…	0.23	503	68	6.41
8	Today kicks off our Active Aging Week! Join us…	0.5	677	116	4.84
9	Plan for healthy living! ?? Learn about the la…	0.54	286	27	9.63
10	Call now to reserve your space in this helpful…	0.5	431	69	5.26
11	Join us this week for a free webinar to learn…	0.35	286	47	5.11
12	Did you catch the latest edition of our newsle…	0.24	1172	107	9.96
13	Join us tomorrow for a free webinar to learn a…	0.31	247	39	5.36

## Data Availability

Datasets are available in a public repository at Mendeley Data, Active Aging, available online: https://data.mendeley.com/drafts/2ynzd8ywcz?folder=0da6c569-15fc-4e3e-b62b-77a62e930451 (accessed on 19 January 2022).

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
