# Peer review of "eHealth Engagement on Facebook during COVID-19: Simplistic Computational Data Analysis"

_ijerph, 2022, doi:10.3390/ijerph19084615_

Round 1

Reviewer 1 Report

The study tries to reveal how social media users viewed health advise on Facebook Pages and Groups with a specific period. The overall level of the paper is good: even if it is quite simple, it is well written, and some important considerations are highlighted.

The scope of the analysis was generally limited and might need to be expanded to increase the applicability of their findings – for instance comparing the eHealth-engagement on Facebook during COVID-19 with another social media platform like Twitter.

The relevance of the results and how it does translate to better eHealth- engagement should be added in the discussion section.

Although the authors raised important questions on the validity of the study but says nothing about the strength of the study, or whether the research question was important or what new knowledge is contributed.

Author Response

We would like to thank the reviewer for the constructive comments, which we feel did well to enhance the manuscript

The scope of the analysis was generally limited and might need to be expanded to increase the applicability of their findings – for instance comparing the eHealth-engagement on Facebook during COVID-19 with another social media platform like Twitter. Whilst CrowdTangle allows access to Instagram, Reddit and Twitter, there were challenges with regards to including data from these social network sites. Both Instagram and Reddit data did not contain all the elements used by Facebook and we felt that important information would be lost. Twitter, through CrowdTangle only allows for 7 days back data and not a full year, so that was excluded too. A statement has been included in the new manuscript.

The relevance of the results and how it does translate to better eHealth- engagement should be added in the discussion section. Thank you for highlighting this. It is an important element, which no includes reference to how online behaviour translates into offline behaviour in the new manuscript.

Although the authors raised important questions on the validity of the study but says nothing about the strength of the study, or whether the research question was important or what new knowledge is contributed.  Several sentences referring to the strength of the study and new knowledge has been included in the Discussion and Conclusion of the manuscript.

Reviewer 2 Report

  1. The title really captures the content of the article. It is a simplistic computational data analysis (2-3). An important problem, because e-health during closure, or restriction of mobility is essential for all. The authors posed a donry research problem: "how social media users viewed health advice on Facebook Pages and Groups" (26-27). It is good to clarify the research question "The aim of the study was how users engaged, what their sentiments were and if any themes were detectable from the content of messages" (28-29). However, I don't understand why the unfortunate statement "Ultimately" is used. "Ultimately, social media can be considered to reach large groups of people and spread content on health issues" (29-31). The role of social media today is undeniable, including in the health field, and especially during the COVID-19 pandemic.
  1. Mark Zuckerberg revealed more plans for Facebook's growth in 2021. The company wants to focus on younger users, because young people definitely prefer TikTok and Snapchat. Facebook presented to its shareholders a very disturbing report. Young people are gradually leaving the company. The social media giant has been reporting clear declines in the share of teens and young twenty-somethings among its users for over 3 years now. There are several reasons for this, of course. That's something to keep in mind, and it's good that the focus here is on the problems of older people. The cited studies from Canada may change dynamically (50-51). It's worth paying attention in your research to this trend, which can take many forms.
  2. When using CrowdTangle it is worth mentioning that it is connected to Facebook, or am I mistaken. The statement in the methodology that "This study employed third party software to collect social media updates on Facebook groups and pages" (96-97) needs clarification. To be clear - the program is not limited to Facebook activities only. It covers other portals such as Instagram, Twitter and Reddit. To run it, you just need to install a free extension in your Google Chrome browser. What sets CrowdTangle apart from other social media competition checking software is free access to information such as: link sharing frequency; number of profiles sharing a post; social media interactions; testimonials, comments and shares over a specified time period. The use of this tool is clearly described in this article, although it is worth mentioning that not only could you compare other social media, but also show the weaknesses of analyzing just one social media.
  1. I am a little lacking in a more thorough review of the research on the issue under study. E-health is a new and important issue. The presence of this issue seems in a cursory review of research - quite vividly present. Perhaps a sentence is needed that this is indeed a pioneering, albeit simple study, or that it marks a new area of research. I look forward with hope to a response on this issue.
  2. The results of the study are very well designed. This can be clearly seen in both the description and the graphic design. It is my hope and desire that the authors will take the second step and do the content. It would be a shame not to analyze it after this research. I encourage the authors to do such research, which would certainly expand contemporary knowledge.

I congratulate the authors on the research effort they undertook.

Author Response

We would like to thank the reviewer for their kind and constructive comments and wish to respond as follows:

  1. The title really captures the content of the article. It is a simplistic computational data analysis (2-3). An important problem, because e-health during closure, or restriction of mobility is essential for all. The authors posed a donry research problem: "how social media users viewed health advice on Facebook Pages and Groups" (26-27). It is good to clarify the research question "The aim of the study was how users engaged, what their sentiments were and if any themes were detectable from the content of messages" (28-29). However, I don't understand why the unfortunate statement "Ultimately" is used. "Ultimately, social media can be considered to reach large groups of people and spread content on health issues" (29-31). The role of social media today is undeniable, including in the health field, and especially during the COVID-19 pandemic. Yes, "ultimately" was the wrong word to use and we have changed the sentence, which now reads better (line 30 - 33)
  1. Mark Zuckerberg revealed more plans for Facebook's growth in 2021. The company wants to focus on younger users, because young people definitely prefer TikTok and Snapchat. Facebook presented to its shareholders a very disturbing report. Young people are gradually leaving the company. The social media giant has been reporting clear declines in the share of teens and young twenty-somethings among its users for over 3 years now. There are several reasons for this, of course. That's something to keep in mind, and it's good that the focus here is on the problems of older people. The cited studies from Canada may change dynamically (50-51). It's worth paying attention in your research to this trend, which can take many forms. Thank you for this valuable information which lead to the discovery of other valuable studies which has been inclusions in the Lit Review and Discussion sections. (Line 47 - 57; 63 - 76; 106 - 109; 344 - 347 and 397 - 399)
  2. When using CrowdTangle it is worth mentioning that it is connected to Facebook, or am I mistaken. The statement in the methodology that "This study employed third party software to collect social media updates on Facebook groups and pages" (96-97) needs clarification. To be clear - the program is not limited to Facebook activities only. It covers other portals such as Instagram, Twitter and Reddit. To run it, you just need to install a free extension in your Google Chrome browser. What sets CrowdTangle apart from other social media competition checking software is free access to information such as: link sharing frequency; number of profiles sharing a post; social media interactions; testimonials, comments and shares over a specified time period. The use of this tool is clearly described in this article, although it is worth mentioning that not only could you compare other social media, but also show the weaknesses of analyzing just one social media. You are correct, CrowdTangle is owned by Facebook and a statement has been included (line 49 - 50). The sentence, now on line 131 and 132 has been improved to clarify "third party software" and further in the study a clarification is given as to why only Facebook data was used.
  1. I am a little lacking in a more thorough review of the research on the issue under study. E-health is a new and important issue. The presence of this issue seems in a cursory review of research - quite vividly present. Perhaps a sentence is needed that this is indeed a pioneering, albeit simple study, or that it marks a new area of research. I look forward with hope to a response on this issue. Please see paragraph in line 425 - 431. 
  2. The results of the study are very well designed. This can be clearly seen in both the description and the graphic design. It is my hope and desire that the authors will take the second step and do the content. It would be a shame not to analyze it after this research. I encourage the authors to do such research, which would certainly expand contemporary knowledge. Due to our inexperience in NLP, we are taking the second article with content analyses slower. This content is very valuable and we are working on a second paper. Once the results have been verified we will make a submission to a suitable journal.
  3. Thank you again for your encouragement.

Reviewer 3 Report

This is a very important topic that combines important themes (COVID, internet communicational technology) and a very important methodological approach that includes so-called
big data analyses and visualizations. 

However, the style and level of English language in this manuscript are not sufficient enough for publishing in international journals. 
It is very difficult even to follow the text and to understand what authors want to present to readers. 
That is why, in my opinion, the manuscript of this valuable topic, should be rewritten in accordance with accepted standards in the scientific community. 

Author Response

We thank you for your comments and value your input. We have sent the article for independent language editing and made some additional changes that we hope you will find acceptable.

Round 2

Reviewer 3 Report

In my opinion, the manuscript is still not appropriate for publishing due to many deficiencies in, above all general academic writing skills, which are reflected in posting problems, understanding relations and levels of notions and variables, and in presenting results. The text is full of trivial facts and insights and without any deeper understanding. Therefore my recommendation is expressed in the next section: major revision

Author Response

In my opinion, the manuscript is still not appropriate for publishing due to many deficiencies in, above all general academic writing skills, which are reflected in posting problems, understanding relations and levels of notions and variables, and in presenting results.

The manuscript has been sent to a second editor for clarification of the above. Minor changes were effected to language use, a paragraph was added to the Methodology Section and a paragraph added to imply connectedness in the Discussion Section.

The text is full of trivial facts and insights and without any deeper understanding. Therefore my recommendation is expressed in the next section: major revision

No trivial facts or insights were removed as the paper never intended to have a deeper psychological meaning. The paper focused on basic information, presented simplistically from a computational social science perspective to gauge usefulness in the field of Psychology. Without in-depth analysis of the content (which is earmarked for a separate paper), it cannot be "more deeply" analyse and integrated. The paper is intended for a factual, simplistic digital presentation of the dataset.
